# Adenoviral-Vectored Centralized Consensus Hemagglutinin Vaccine Provides Broad Protection against H2 Influenza a Virus

**DOI:** 10.3390/vaccines10060926

**Published:** 2022-06-10

**Authors:** Erika M. Petro-Turnquist, Brianna L. Bullard, Matthew J. Pekarek, Eric A. Weaver

**Affiliations:** Nebraska Center for Virology, School of Biological Sciences, University of Nebraska-Lincoln, 4240 Fair Street, Lincoln, NE 68504, USA; epetro-turnquist2@huskers.unl.edu (E.M.P.-T.); briabullard@gmail.com (B.L.B.); mpekarek2@huskers.unl.edu (M.J.P.)

**Keywords:** influenza, H2N2, consensus, vaccine, adenovirus, species C

## Abstract

Several influenza pandemics have occurred in the past century, one of which emerged in 1957 from a zoonotic transmission of H2N2 from an avian reservoir into humans. This pandemic caused 2–4 million deaths and circulated until 1968. Since the disappearance of H2N2 from human populations, there has been waning immunity against H2, and this subtype is not currently incorporated into seasonal vaccines. However, H2 influenza remains a pandemic threat due to consistent circulation in avian reservoirs. Here, we describe a method of pandemic preparedness by creating an adenoviral-vectored centralized consensus vaccine design against human H2 influenza. We also assessed the utility of serotype-switching to enhance the protective immune responses seen with homologous prime-boosting strategies. Immunization with an H2 centralized consensus showed a wide breadth of antibody responses after vaccination, protection against challenge with a divergent human H2 strain, and significantly reduced viral load in the lungs after challenge. Further, serotype switching between two species C adenoviruses enhanced protective antibody titers after heterologous boosting. These data support the notion that an adenoviral-vectored H2 centralized consensus vaccine has the ability to provide broadly cross-reactive immune responses to protect against divergent strains of H2 influenza and prepare for a possible pandemic.

## 1. Introduction

In 1957 the H2N2 “Asian flu” pandemic quickly swept the globe, causing an estimated 2 million deaths in the first year alone [1]. This pandemic strain resulted from a zoonotic transmission event from a reassorted H2N2 avian and H1N1 human influenza A virus [2]. H2N2 then circulated in human populations until 1968, when the virus was reassorted again with H3 influenza and resulted in the “Hong Kong” flu. Serological analysis indicates that people under 50 years old have little-to-no immunity to H2N2 [3] and that seasonal influenza vaccination does not provide cross-protection [4]. Importantly, H2 influenza is still consistently isolated from waterfowl, wild birds, and domestic ducks [5,6,7] and exhibits substantial pandemic potential. In 2005, the World Health Organization (WHO) set forth a call to action to create and stockpile vaccines against influenza subtypes with pandemic potential [8,9]. However, the mutability and reassortment tendencies of influenza viruses make predicting and protecting against pre-pandemic strains of influenza challenging [10]. It is imperative that efforts be allocated toward creating a broadly protective vaccine against potential pandemic viruses, including H2 influenza. Recent work characterized the utility of multivalent inactivated virus vaccines (IIV) against H2 [11], a cold-adapted live attenuated influenza virus (LAIV) H2 vaccine [12,13,14,15], and computationally derived consensus vaccines against influenza viruses exhibiting pandemic potential [16,17,18,19,20]. Lenny et al. observed that immunizing with non-human H2 isolates in an IIV platform resulted in moderate antibody responses; however, these responses were low compared to a LAIV H2 vaccine [12]. Nasal administration of the LAIV H2 vaccine was demonstrated to have high immunogenicity in mice and ferrets [14] and has historically been shown to elicit long-lasting systemic cellular immune responses and enhanced levels of secretory IgA at mucosal surfaces [21]. However, clinical translation of the H2N2 LAIV vaccine to humans has remained a challenge [15], and consistency between clinical trials suggested that achieving broad heterologous protection may require a multivalent formulation [12,14]. Further, this vaccine modality has the potential to mutate [22] and reassort with circulating strains during co-infection [23,24]. Although reassortment between LAIV and circulating strains has not yet been described in humans, this was recently observed in swine populations after vaccination with the nonstructural protein 1 (NS1)-truncated commercial LAIV vaccine [25]. The epidemiological repercussions of this reassortment event have yet to be elucidated, but this highlights the necessary precautions that need to be considered when administering LAIV vaccines. Importantly, the AA ca virus, initially created and characterized in the late 1960s [26,27], has been used as the backbone of seasonal LAIV administered in the United States. The attenuation and utility of AA ca has been an invaluable contribution to increasing our understanding of immunological responses to LAIV vaccines and is still used as the master donor virus (MDV) of many LAIV vaccines today, highlighting its importance in vaccine development.

The inherent immunogenicity and enhanced stimulation of innate and adaptive immune responses make adenoviral-vectored vaccines an attractive platform to deliver antigens in vivo [28]. Unsurprisingly, adenoviruses (Ads) are the leading vectors applied in gene therapy clinical trials to treat cancer, cardiovascular diseases, and various infectious diseases [29]. The utility of Ad as a viral vector against measles, hepatitis B and C, rabies, Ebola, anthrax, human immunodeficiency virus-1 (HIV-1), malaria, and influenza was previously described [30], and currently, all viral-vectored vaccines approved for use against the ongoing SARS-CoV-2 pandemic are based on Ad vectors [31,32,33,34,35]. Ad has a stable, non-integrative double-stranded DNA genome with a large packaging capacity that can be easily manipulated to be rendered replication-incompetent, exhibiting an enhanced safety profile in comparison to LAIV. Further, the manufacturing of Ad can be easily scaled up to mass-produce large quantities of vaccines in the event of a pandemic outbreak [36]. Finally, Ads are grown entirely in cell culture, mitigating the induction of egg-related allergy responses seen with seasonal influenza virus vaccines [28]. For these reasons, we utilized a replication-defective Ad vector to deliver our vaccine immunogen.

In order to address the concern of limited cross-protection after immunization with a wildtype influenza sequence, we utilized a centralized consensus hemagglutinin (HA) design to offer enhanced cross-reactive immune responses. The HA is the most abundant glycoprotein on the surface of influenza and is a central target of neutralizing antibodies during immune responses against vaccination and infection. Our centralized consensus approach uses representative wildtype HA sequences from each major branch to create a final consensus construct that localizes to the central node of the phylogenetic tree. These synthetic constructs minimize the genetic and antigenic differences seen in influenza HA and remove sampling and reporting biases that naturally occur with global influenza surveillance programs. We previously showed our centralized consensus approach to effectively enhance protection against H1, H3, and H5 influenza A viruses and found that these synthetic immunogens outperformed wildtype strains [16,17]. Further, immunization with a centralized consensus construct of the envelope (env) protein of HIV-1 demonstrated higher efficacy against a wide breadth of divergent strains compared to wildtype isolates [37,38,39]. Because of this previous success, we applied this strategy to create a broadly protective consensus H2 (H2-Con) influenza vaccine.

Here, we describe a method of pandemic preparedness by creating an adenoviral-vectored centralized consensus hemagglutinin construct against previously circulating pandemic strains of human H2 influenza A virus. We found that H2-Con had the ability to induce broadly protective antibody titers when delivered first by Ad6 and boosted with Ad5. Further, we observed complete protection against challenges with a divergent H2 strain in vivo and an enhanced capacity to clear the virus from the lungs. Based on our results, we showed that an Ad-vectored vaccine strategy could be used to prepare for an impending pandemic and warrants further investigation into clinical applications.

## 2. Materials and Methods

### 2.1. Ethics Statement

All protocols were reviewed and approved by the Institutional Biosafety Committee (IBC) at the University of Nebraska–Lincoln, as specified in the IBC protocol 1908. Six- to eight-week-old female BALB/c mice were purchased from Jackson Laboratory (Bar Harbor, ME, USA) and housed at the Life Sciences Annex building on the University of Nebraska–Lincoln (UNL) campus under the Association for Assessment and Accreditation of Laboratory Animal Care International (AAALAC) guidelines. All procedures involving infectious human H2 influenza were completed in the Nebraska Veterinary Diagnostic Laboratory (NVDL) BSL-3/ABSL-3 facility at UNL. The protocols were approved by the UNL Institutional Animal Care and Use Committee (IACUC) (Project ID 1908). All animal experiments were carried out according to the provisions of the Animal Welfare Act, PHS Animal Welfare Policy, the principles of the NIH Guide for the Care and Use of Laboratory Animals, and the policies and procedures of UNL. All immunization, bleeds, and infections were completed under either isoflurane or ketamine/xylazine-induced anesthesia. After the lethal influenza challenge, mice that reached 25% weight loss were humanely euthanized.

### 2.2. Influenza Viruses

The following reagents were obtained through BEI Resources, NIAID, NIH: A/Singapore/1/1957 (Singapore/57) [NR-31132], A/Japan/305/1957 (Japan/57) [NR-347], A/Formosa/313/1957 (Formosa/57) [NR-15561], A/Korea/426/1968 (HA, NA) × A/Puerto Rico/8/1934 (Korea/68) [NR-3527], A/Rockefeller Institute/5/1957 (RI/57) [NR-3525], A/Japan/305/1957 (Japan/57) [NR-3171], and A/Taiwan/1/1964 (Taiwan/64) [NR-3173]. All viruses were grown in specific pathogen-free (SPF) eggs, and the allantoic fluid was stored at −80 °C. Grown viruses were quantified by standard hemagglutination assay (HAU).

### 2.3. Design of Centralized Consensus Hemagglutinin and Phylogenetic Analysis

All unique, full-length human H2 hemagglutinin (HA) sequences were downloaded from the Genbank as of 15 June 2007. The resulting 98 sequences were aligned with ClustalW and used to create a phylogenetic tree. Sixteen H2 HA sequences were chosen to represent the major branches of the phylogenetic tree and used to create a synthetic centralized consensus protein sequence (accessions: AY209979, AY209957, CY125854, CY014976, AY209954, L11134, L20409, L11142, AY209959, AY209963, AY209961, AY209962, L11133, L11125, L11126, and D13579). An alignment of the 16 representative sequences and H2-Con is shown in Appendix A. Briefly, the 16 sequences were aligned using ClustalW, and the most common amino acid at each position was determined and incorporated into the synthetic protein sequence. This sequence was then realigned to all unique human H2 strains, and a Neighbor-joining phylogenetic tree was constructed to visualize the designed centralized consensus protein using Geneious 11.1.5 software and a Jukes–Cantor model with a Blosum62 cost matrix. 

### 2.4. Construction of Recombinant Adenoviruses

The H2 centralized consensus was codon-optimized for mammalian expression and synthesized by Genscript, Inc. This consensus H2 construct was previously described and published by Lingel A. et al. [17]. In this study, we included a more detailed and comprehensive characterization of the H2-con immunogen. The consensus construct was cloned into first-generation replication-defective Ad5 (E1/E3 deleted) or Ad6 (E1 deleted) vectors, as previously described [40]. Recombinant Ad5 and Ad6 carrying the H2-Con gene were linearized by PacI (New England Biolabs) and transfected into human embryonic kidney 293 cells (HEK293) using PolyFect Transfection Reagent (Qiagen). The rescued constructs were observed for cytopathic effects (CPE) and then amplified in the human embryonic kidney (HEK) 293 cells by sequential passaging until a final infection of a 10-cell stack cell factory (Corning). Ad5-H2-Con and Ad6-H2-Con were purified by two steps of cesium chloride ultracentrifugation, then desalted using Econo-Pac 10DG Desalting Columns (Bio-Rad). Final vector preparations were quantified by OD260, and infectious units were confirmed to be equal using the AdenoX Rapid Titer kit according to standard manufacturer’s instructions (Clontech Laboratories, Mountain View, CA, USA).

### 2.5. Western Blot

Protein expression of the Ad5 and Ad6 constructs were confirmed with Western blot analysis. Confluent HEK293 cells were infected at an MOI of 500, incubated for 48 h at 37 °C, 5% CO_2_, then harvested. Cells were washed one time with 50 µL of DPBS, then denatured in 2X Laemmli and 2-mercaptoethanol buffer and heated at 100 °C for 10 min. Samples were passed through a Qiashredder (Qiagen) and run on 12.5% SDS-PAGE. Protein was transferred to a nitrocellulose membrane, blocked in 5% milk in 1X TBST for 1 h, then probed at 4 °C overnight with g-α-A/Singapore/1/1957 (NR-3150) at 1:1000 dilution or m-α-GAPDH-HRP (sc-47724) at a 1:500 dilution in 1% milk-1X TBST. The membrane was washed twice with 1X TBST and then probed with a 1:2000 dilution of secondary donkey α-goat-HRP antibody in 1% milk-1X TBST for 1 h. The final membrane was washed twice with 1X TBST and developed with SuperSignal West Pico Chemiluminescent Substrate (Thermo Scientific, Waltham, MA, USA).

### 2.6. Vaccination

Groups of female BALB/c mice (*n* = 5) were immunized via intramuscular injection of 10^10^ vp of either Ad5- or Ad6-H2-Con diluted in 50 uL of DPBS. Two 25 μL doses were delivered to each hind leg for immunization. Three weeks post-prime, mice were bled via submandibular puncture and boosted with homologous or heterologous adenoviral-vectored (Ad5/Ad6)-H2-Con vaccine. Two weeks later, the mice were bled via cardiac puncture under ketamine and xylazine-induced anesthesia. Serum was separated from whole blood using a BD Microtainer Blood Collection Tube (Becton Dickinson) and centrifuged at 6800× *g* for 2 min, then used in subsequent analysis.

### 2.7. Hemagglutination Inhibition (HI) Assay

Sera from vaccinated mice were treated at 37 °C overnight with receptor destroying enzyme (RDE; Denka Seiken) in a 1:3 ratio (sera: RDE), then heat-inactivated at 56 °C for 30 min. Sera were diluted to a final concentration of 1:10 in DPBS, and 25 μL of sera was serially diluted two-fold down a 96-well v-bottom plate. V-bottom plates were used to be able to visualize red blood cell pellets during development. An amount of 25 µL of four hemagglutinating units (4HAU) was added to all wells, and plates were incubated at room temperature for 1 h before adding 50 µL of 0.5% chicken red blood cells (cRBCs). Agglutination patterns were read after 30 min of incubation at room temperature. 

### 2.8. Lethal Influenza Challenge, Tissue Collection, and Lung Viral Titers

Groups of female BALB/c mice (*n* = 10) were primed with 10^10^ viral particles (vp) of Ad5-H2-Con, Ad6-H2-Con, or DPBS, then boosted three weeks later with a homologous or heterologous vaccine. Two weeks after boosting, all mice were challenged with 20 times the median lethal dose (20MLD_50_) of A/Korea/426/1968 and humanely euthanized at ≥25% baseline weight loss. This virus was chosen from a preliminary lethality study (Appendix A) and was already pathogenic in mice without the need for mouse adaptation. At three days post-infection (3 dpi), half of the mice (*n* = 5 mice/group) were sacrificed, and lungs were collected for analysis of lung viral titers by qPCR. Lungs were homogenized in DPBS, centrifuged at max speed for 10 min, then 100 µL of the supernatant was subjected to Purelink viral RNA/DNA extraction (Invitrogen) according to manufacturer’s instructions. Real time-qPCR was performed using the Luna Universal One-Step RT-qPCR Kit (NEB) and the Universal influenza A primer set (BEI Resources, NR-15579 and NR-15580) with the following cycling parameters: 55 °C for 10 min, 95 °C for 2 min, and 45 cycles of 95 °C for 15 secs and 60 °C for 1 min. Final viral RNA concentrations were determined based on a standard curve from a known quantity of A/Puerto Rico/8/34 viral RNA. 

### 2.9. Statistical Analysis

All data were analyzed using GraphPad Prism software. Data are expressed as the mean with standard error (SEM). One-way ANOVA statistical analyses were used to compare groups in HI titers and lung viral titers. A *p*-value of <0.05 was considered statistically significant (* *p* < 0.05; ** *p* < 0.01; *** *p* < 0.001; **** *p* < 0.0001).

## 3. Results

### 3.1. Construction and Characterization of Centralized Consensus H2 Vaccine Immunogen

The current vaccination strategies incorporate inactivated wildtype strains into clinical vaccines. In 1957 this was the tactic used to immunize against circulating H2N2 influenzas. However, this method induced strain-specific responses and provided limited protection as the pandemic progressed over a ten-year period [2,10]. In order to evaluate the ability to enhance cross-protection, we generated a centralized consensus hemagglutinin protein based on antigenically representative human H2 influenza A viruses from 1957 to 1968. This approach produces a vaccine immunogen that is equidistant to all other divergent strains and has lower sequence divergence than any given wildtype strain. Further, this technique removes sampling bias if testing locations have more thorough surveillance and sequencing methods compared to other locations. As expected, phylogenetic analysis revealed that this synthetic immunogen localized to the central node of the human H2 influenza A virus tree (Figure 1A). H2-Con was then cloned into a replication-defective human adenovirus type 5 or human adenovirus type 6 viral vector, and hereafter are referred to as Ad5-H2-Con or Ad6-H2-Con. The protein expression of the synthetic H2 immunogen from each adenovirus was confirmed by Western blot (Figure 1B), and antigenic relatedness to strains used in the study was determined (Figure 1C). The strains used in this study were determined based on reagent availability and sequence divergence from the H2-Con immunogen. We included these sequences in determining if H2-Con could induce protective responses against highly divergent strains that were circulating in both the 1950s and the 1960s.

### 3.2. Adenoviral-Vectored Consensus H2 Immunogen Induces Protective Humoral Immune Responses

We first evaluated humoral immune responses after a single vaccination with either Ad5-H2-Con or Ad6-H2-Con and compared the responses to a PBS sham negative control group. The breadth of antibody responses was assessed using six different human H2 influenza A viruses selected from 1957 to 1968 and assayed by hemagglutination inhibition (HI) titer. HI titers ≥ 40 are considered to correlate to 50% protection [41] and are the current standard measurement of vaccine efficacy in the field. Immunization with either Ad5- or Ad6-H2-Con induced equal levels of HI antibodies after a single immunization among all six viruses tested (Figure 2). Ad5-H2-Con induced protective antibody titers of ≥40 against Korea/68, Singapore/57, and Taiwan/64. Ad6-H2-Con exhibited protective titers against Singapore/57, Taiwan/64, and Japan/62. There were no statistically significant differences between the Ad5- and Ad6-H2-Con immunization groups, suggesting that a single vaccination with either Ad5 or Ad6 will induce equal levels of HI antibody responses.

Next, we assessed protective antibody titers after either a homologous or heterologous boost immunization regimen. A common concern with viral-vectored vaccines is immune-mediated neutralization of the carrier vector before mounting a significant immune response against the delivered immunogen. In light of this, we evaluated HI titers against the same panel of human H2 isolates after either a homologous or heterologous boost vaccination with a serotype switched viral vector carrying the H2 centralized consensus immunogen (Figure 3). Immunization with Ad5/Ad5-H2-Con and Ad6/Ad6-H2-Con homologous vaccine strategy induced protective titers of ≥40 against four out of six human H2 isolates (67% response). In contrast, serotype switching between Ad5/Ad6-H2-Con and Ad6/Ad5-H2-Con induced an average protective HI titer of ≥40 against five out of six (83% response) and six out of six (100% response) human H2 isolates, respectively. We observed a trend of Ad6/Ad5-H2-Con eliciting an average HI titer of ≥40 against 100% of the strains included in our panel; however, it is worth noting that the differences in HI titer between the groups against Japan/57 did not reach statistical significance (Figure 3E). Additionally, serotype switching resulted in significantly higher HI antibody titers compared to homologous prime-boost immunization (Figure 3). These data indicate that serotype switching prime-boost strategies can elicit protective antibody titers against a wider range of human H2 isolates and a higher magnitude of protective HI antibody titers as compared to a homologous prime-boost strategy.

### 3.3. H2 Consensus Provides Protection against Lethal A/Korea/426/1968 Challenge

After completing immune correlate studies, we tested if the H2-Con vaccine is capable of eliciting protection against a divergent lethal H2 influenza infection. BALB/c mice (*n* = 10) were infected with 20 times the median lethal dose (20MLD_50_) of Korea/68 after a prime-boost vaccination. Mice were vaccinated with homologous or heterologous Ad5/6-H2-Con or a PBS mock vaccination. Five mice per group were observed for 14 days post-infection (dpi) to assess differences in morbidity and mortality. We saw complete protection from lethal H2 challenge in all groups vaccinated with the H2-Con vaccine regardless of serotype switching (Figure 4). No appreciable differences in weight loss were observed between any of the adenoviral-vectored H2-Con groups, while mock vaccinated mice quickly lost weight starting at 3 dpi (Figure 4A). Half of the mice (*n* = 5) were sacrificed 3 dpi to examine the effect of vaccination on lung viral titers after the challenge. We saw significantly reduced levels of viral RNA copies present in the lungs in all vaccinated groups compared to the PBS negative control group (Figure 4B). All unvaccinated animals succumbed to challenge by 7 dpi, while all H2-Con vaccinated mice survived over the two-week study (Figure 4C).

## 4. Discussion

Though H2 influenza has not circulated in humans since 1968, it exhibits significant pandemic potential due to a lack of immunity in a large proportion of the population [3], no cross-protection from seasonal vaccination [4], and prominent avian reservoirs [5,6,7]. Because of this, we explored the ability to provide broad protection against pandemic strains of human H2 influenza by utilizing a centralized consensus immunogen delivered by an adenoviral vector. We previously showed that vaccinating with centralized consensus H1, H3, and H5 influenza strains provides a greater breadth of immunity against divergent strains than wildtype immunogens in vivo [16,17]. Here we expanded our findings to the human H2 influenza A virus.

We first noted that priming with either Ad5 or Ad6 induced similar levels of HI antibody titers, with no statistically significant differences seen between the two vaccine regimens after one dose, suggesting that a single immunization may provide protective HI antibody titers regardless of the serotype. We were surprised to see the lowest HI antibody titers against Japan/57 and RI/57, despite these strains having the highest percent sequence identity to the H2-Con construct (98.58% and 98.04%, respectively). A closer analysis of these sequences revealed a shared alanine-to-threonine mutation in both Japan/57 and RI/57 that was not included in the H2-Con construct and is not present in the rest of our panel of representative divergent strains (Appendix A). Importantly, the positioning of this amino acid substitution is located proximal to antigenic sites B and D of the H2 hemagglutinin [42,43,44] and an asparagine residue. It is possible that these two viruses possess an additional “masking” glycosylation site that would result in decreased antibody titers measured by HI. Indeed, this mechanism of viral escape was previously described in the context of two other avian-origin influenza viruses: H7N9 [45] and H5N1 [46,47]. Although HI titers are a standard correlate of protection, more precise assays such as ELISAs B cell epitope mapping could help elucidate additional antibody-mediated effector functions against these strains. Further studies assessing the implications of this residue can provide insight into immune escape mechanisms employed by H2 influenza A viruses. Another interesting observation was that priming with Ad6-H2-Con and boosting with Ad5-H2-Con was able to elicit an average protective antibody titer of ≥40 against 100% of the strains tested. Further, H2-Con provided complete protection against morbidity and mortality after a challenge with 20 times the median lethal dose of a divergent human H2 strain. Complete protection against morbidity was likely attributed to enhanced viral clearance, as mice immunized with H2-Con had significantly reduced the presence of the virus in the lungs by 3 days post-infection. 

A common concern with using an adenoviral-vectored vaccine platform is the neutralization of the vector after repeated immunization. In light of this, we tested the utility of serotype switching between two species C Ad vectors (Ad5 and Ad6). Both Ad5 and Ad6 are very well characterized in several strains of mice, with known receptors, tissue tropism, and immunogenicity. Though Ad5 and Ad6 are both species C adenoviruses, Ad6 exhibits subtle structural differences [48,49,50,51], has lower seroprevalence than Ad5 [52], and can escape preexisting immunity towards Ad5 [53], making it an ideal candidate to be included in an alternative prime-boosting vaccine strategy. The differences in protective HI titers seen between priming with either Ad5- or Ad6- before heterologous boosting are likely due to the increased immunogenicity of Ad6 compared to Ad5. Ad6 was described to efficiently circumvent Kupffer cell sequestration after systemic delivery [54,55] and has enhanced hepatocyte transduction during immunization [49,55]. Because Ad6 is able to efficiently evade this method of sequestration, it is possible this can enhance the protective responses seen in vivo. It is worth noting that, though we saw enhanced antibody titers after heterologous boosting, this did not correlate to decreased protection after the challenge, as all groups immunized with the H2-Con had complete protection after Korea/68 infection. This was not unexpected, as all groups exhibited protective antibody titers against this virus, regardless of heterologous or homologous boosting. In future studies, we would like to expand our challenge virus panel to incorporate viruses that exhibit higher sequence divergence than the virus included in our challenge model. However, the biocontainment practices required for work with infectious human H2 influenza [43] remain a major hurdle to obtaining viruses that are readily lethal in mouse models and the expertise required to complete vaccine studies against avian-origin influenza A viruses. Additionally, given that H2 is still consistently detected in birds, an important contribution to increasing pandemic preparedness would be creating a consensus construct against avian H2 influenzas that would provide rapid and broad protection in the event of a zoonotic transmission event from birds to humans. This could be complemented by also exploring alternative Ad vectors. The approval of Ad-vectored vaccines for use against SARS-CoV-2 has allowed enhanced research of this vaccine modality in recent years, with substantial improvements to the rate of development, identification of protective immune correlates, and new recombinant technologies being identified. Importantly, recent research has begun defining the induction of mucosal immune responses after intranasal administration of recombinant, replication-competent Ad4 expressing a wildtype H5 human influenza [56]. Clinical translation into humans was marked by durable nasal IgG and IgA and enhanced levels of circulating H5 HA-specific CD4+ and CD8+ T cells present in the peripheral blood. Serum neutralizing antibodies could be detected for 26 weeks after immunization, and adverse effects were mild and correlated with peak viral shedding at 6 days post-vaccination [57]. Intranasal delivery of a consensus vaccine construct in a replication-competent Ad vector could offer enhanced mucosal and systemic antibody responses, as well as cellular-mediated immune responses similar to LAIV vaccines but with an enhanced safety profile. To the best of our knowledge, no studies have directly compared the protective efficacy of LAIV versus Ad vectored vaccines and would be an interesting focal point of a future study.

Furthermore, swine are important reservoirs that have been described to facilitate efficient reassortment events between multiple strains of the influenza A virus [58,59]. In 2006, H2 influenza was isolated from two independent swine farms [60], highlighting the potential for these intermediate hosts to enable pandemic strains of H2 to enter human populations. Because of this, it may be beneficial to produce and administer a broadly protective avian H2 influenza virus vaccine to these vulnerable populations and, ultimately, help protect against zoonotic transmission to human populations.

## 5. Conclusions

In this study, we demonstrated the ability of an adenoviral-vectored H2 centralized consensus immunogen to provide broad protection in vivo. H2 influenza exhibits significant pandemic potential, and we addressed this concern by creating a broadly cross-protective vaccine with the ability to protect against a variety of divergent H2 influenza strains. This strategy of pandemic preparedness has the potential to prevent high rates of mortality in the event of another pandemic.

## Figures and Tables

**Figure 1 vaccines-10-00926-f001:**
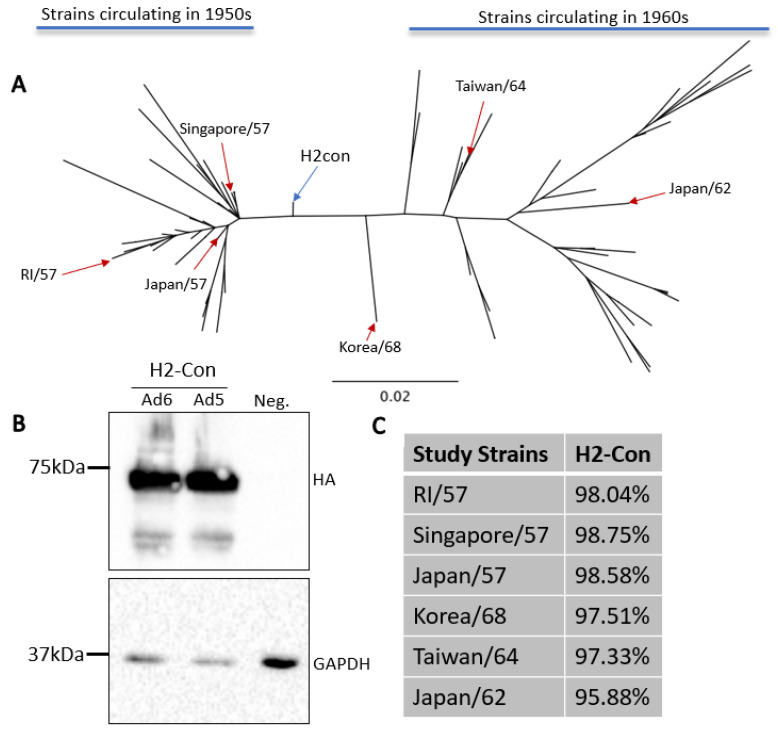
Characterization of centralized consensus human H2 immunogen and study strains. (**A**) The genetic relationships between the centralized consensus and wildtype H2 influenza A virus strains are shown in an unrooted neighbor-joining phylogenetic tree. H2-Con (blue arrow) localizes to the center of all human H2 strains isolated between 1957 and 1968. Reference strains isolated from 1957 to 1964 (red arrows) and used in this study are labeled on the tree. (**B**) Western blot analysis of H2-Con (~72 kDa) in Ad6 and Ad5 viral vectors from two independent experiments. GAPDH (~33 kDa) was probed as a cellular loading control. (**C**) Percent sequence similarity of the wildtype strains used in the study compared to H2-Con.

**Figure 2 vaccines-10-00926-f002:**
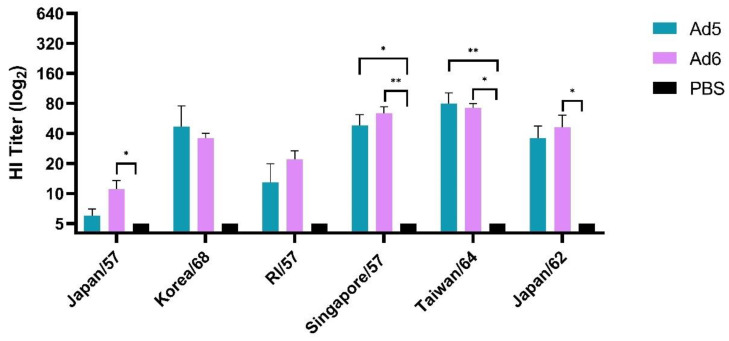
Hemagglutinin inhibition (HI) titers after prime vaccination. BALB/c mice (*n* = 5) were immunized with 10^10^ viral particles (vp) of Ad5-H2-Con, Ad6-H2-Con, or DPBS. Sera from immunized animals were assayed for HI antibody titers against a range of divergent human H2 influenza A viruses. Data are represented as the mean HI titer with standard error (SEM). (*n* = 5; statistical analysis based on one-way ANOVA with Tukey multiple comparison; * *p* < 0.05, ** *p* < 0.01).

**Figure 3 vaccines-10-00926-f003:**
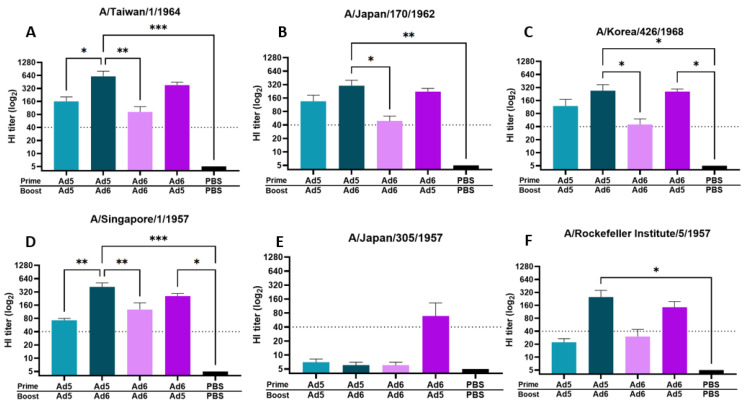
Hemagglutinin inhibition (HI) titers after boost vaccination with homologous or heterologous vaccine regimen. BALB/c mice (*n* = 5) were immunized with 10^10^ vp of Ad5-H2-Con or Ad6-H2-Con and boosted with their homologous or heterologous adenovirus serotype. HI titers against Taiwan/64 (**A**), Japan/62 (**B**), Korea/68 (**C**), Singapore/57 (**D**), Japan/57 (**E**), and RI/57 (**F**) after homologous or heterologous boosting are shown. Data are represented as the mean HI titer with standard error (SEM). The dotted line indicates a protective titer of ≥ 1:40 (*n* = 5; one-way ANOVA with Tukey multiple comparison; * *p* < 0.05, ** *p* < 0.01, *** *p* < 0.005).

**Figure 4 vaccines-10-00926-f004:**
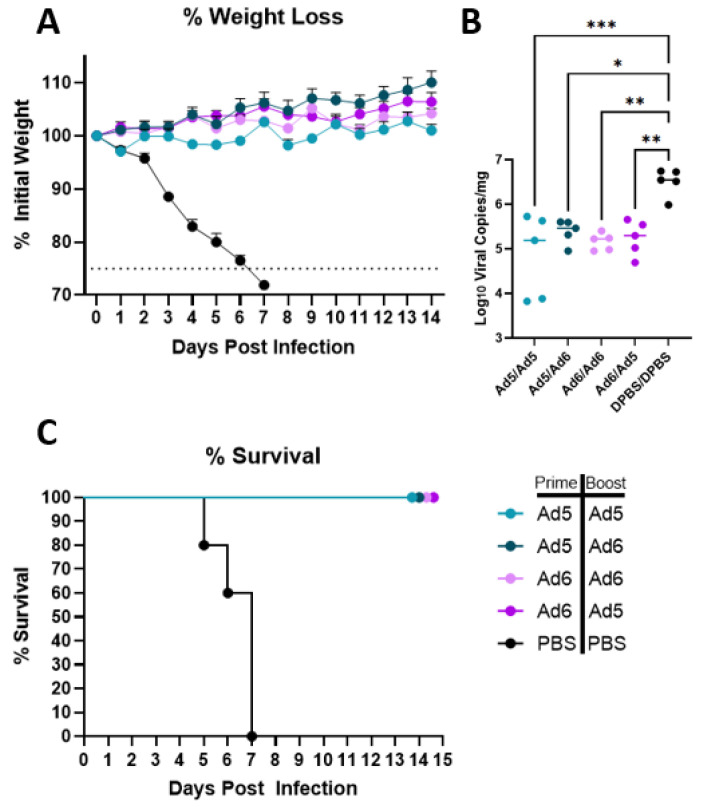
Protection against challenge with a lethal human H2 isolate. BALB/c mice (*n* = 10) were vaccinated with 10^10^ vp of Ad5-H2-Con, Ad6-H2-Con, or PBS and boosted with either homologous or heterologous vaccine regimen. Mice were challenged with 20MLD_50_ of A/Korea/426/1968, then monitored for morbidity (**A**), clearance of viral RNA from the lungs (**B**), and mortality (**C**). Mice that lost 25% of their initial weight were humanely euthanized (*n* = 5; one-way ANOVA with Tukey multiple comparison; * *p* < 0.05, ** *p* < 0.01, *** *p* < 0.005).

## Data Availability

All sequences used to create the consensus immunogen is freely available through the Influenza Research Database at https://www.fludb.org/brc/home.spg?decorator=influenza (accessed on 11 April 2022). All other relevant data will be provided by the corresponding author upon request.

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
