# Peer review of "Adenoviral-Vectored Centralized Consensus Hemagglutinin Vaccine Provides Broad Protection against H2 Influenza a Virus"

_vaccines, 2022, doi:10.3390/vaccines10060926_

Round 1

Reviewer 1 Report

In this short manuscript by Petro-Turnquist et al the authors describe the production of a consensus vaccine against influenza H2. Use is made of adenoviruses  to deliver the the immunogen. The study seems to have been carefully carried out and the results appear convincing. I have a few minor criticisms which should be addressed.

The authors should provide the amino acid sequences of the the consensus HA sequence and those of the 16 viruses used in the alignment. This could be presented in supplementary/supporting information.

There are results which are worthy of comment which the authors do not discuss. In Figure 2 the HI titer against Japan/57 is appreciably lower than for the other viruses. Also the response for the Ad5 variant is much lower than for Ad6. These observations should be considered.

Similarly in Figure 3E the authors should suggest reasons why the Ad6/Ad5 vaccine regimen produces a high HI titer whereas none of the others have an effect. Along the same lines, the authors could suggest why Ad6/Ad5 has a remarkably unique effect against A/Japan305/1957 whereas for all the other viruses examined the 4 different regimens gave essentially similar results.

I understand the reasons for using 2 serotype C adenoviruses but wouldn't it have been more interesting to use Ad5 and a virus from a different serotype?

This is a reasonably interesting study which would be helped by more careful and extensive discussion of the data presented.

Reviewer 2 Report

The manuscript describes the development of an adenovirus based vaccine that elicits broad protection against H2 influenza A viruses.

To achieve the broad protection, a H2-con antigen was constructed based on the sequence of several H2 strains known for infecting human. This broad antigen was cloned into the Ad5 and Ad6 vectors. The Ad5-H2-con or the Ad6-H2-con vaccines were immunized twice i.m. Homologous and heterologous vaccination protocols were performed and the antibody response was studied by comparing the HI titers. A challenge with the strain Korea/68 was performed on the immunized animals.

Comments

1-The main novelty of the manuscript is related to the H2-con antigen. However, it is not possible to judge the efficacy of the H2-con antigen because it is not compared with at least one H2 WT antigen that should have been cloned in the same viral vectors.

2-Fig, 3E. This fig reveals that the HI titers of the Ad6/Ad5 vaccine in not significantly different to the others. Therefore, it cannot be concluded that the Ad6 /Ad5 vaccine is efficient against 100% of the strains tested.

3-It was surprising to see the weak neutralization potential of the Ad/Ad5 vaccine against the Japan/57 strain (Fig. 3E). even if the HA protein of this virus is 98,58% identical to the H2-con. Interestingly, the HI titers induced by antibodies generated by the Ad6/Ad5 vaccine were more efficient against the Japan/62 strain that is more divergent (95,88%) from the H2-con.

Tis is an interesting result and it should be discussed in the manuscript. An amino acid alignment between all the strains showed at Fig1C with the H2-con should reveal the a.a. that are potentially involved in this 'break' of neutralization. An attempt to identify this region could be very interesting.

4-A challenge experiment using the Japan/57 should be performed to confirm that the Ad6/Ad5, and not the Ad6/Ad5 vaccination regimen is capable to induce protection. As expected, the chalenge with Korea/68 generated 1000% protection with all the formulation tested which, is not very informative because the challenge could not discriminate between the different vaccines. 

Reviewer 3 Report

The authors presented an interesting, well-written work attracted by its clear design and clarity of presentation. However, after reading the article, I have a few comments and would like the authors not only to respond to these comments in a rebuttal letter, but also to make appropriate corrections to the text.

Point 1: There is a cold-adapted attenuated virus, A/Ann Arbor/6/60 (H2N2) (A/Singapore/1/57-like virus), which was prepared in the 1960s by Dr. Maassab and is currently being used in the preparation of MedImmune reassortant LAIV as a source of internal genes. Could you please explain what are the advantages of your vaccine over a strain that, if H2N2 viruses are back in circulation, could be used as a LAIV?

Point 2: Line 47-48. The authors stated that LAIV “have the potential to mutate and reassort during co-infection [15, 16], introducing additional risk.” However, this is a speculative statement.A similar phenomenon can and does occur when two epidemic viruses co-infect, but no one has yet been able to prove that LIV can naturally reassort with an epidemic virus. There is no evidence of reassortment of LAIV with human influenza viruses. References 15-16 merely state the major concern that such a situation is possible.

Point 3: line 49-50. The authors stated that clinical translation [of LAIV] has remained a challenge due to reduced antibody responses [12]. However, it is well known that there is no correlation between humoral immune response to LAIV and LAIV efficacy. The major criterium of the quality of LAIV is cell-mediated response.

Point 4: Line 50-51. The authors mentioned poor replication of LAIV in humans [18]. However, the level of replication of cold-adapted vaccine strains is quite enough to evoke an adequate immune response in a person. In addition, according to the authors [18], “…the H2N2 LAIV was highly infectious…”

Point 5: Figure 1A. I believe that it is necessary to add such a key virus for LAIV as A/Ann Arbor/6/60 (H2N2).

Point 6: Line 306-307. It seems to me that it would be appropriate to draw an analogy with anti-COVID-19 vaccines based on the adenovirus vectors. You partially did this in the Introduction (Line 59), but it is also very relevant in the Discussion.

Round 2

Reviewer 2 Report

Dear Authors,

I have made 4 comments concerning this manuscript and I accept your responses for 3 out of 4. 

However, I disagree with your response concerning the following comment:

1-The main novelty of the manuscript is related to the H2-con antigen. However, it is not possible to judge the efficacy of the H2-con because it is not compared with at least one H2 WT antigen that should have been cloned in the same vector.

You claim that a consensus sequence will always be superior to a WT sequence based on your past demonstration with a consensus approach for the H1 subclass (Plos One) and the H1, H3 and H5 subclass (Sci Rep). You also claim that the demonstration that a consensus sequence with the HIV-env provide further support and justify that you do not need a control in the current study. 

Response:

-To me the reference to the HIV work do not provide a lot of support since it is a completely different virus.

-The consensus sequence of the Sci Rep study (ref 17) also include the H2 subgroup of influenza virus. The demonstration for protection were performed using the H1, H3 and the H5 strain. Therefore, I am wondering if you have been using the same consensus protein presented  in Sci Rep paper for this manuscript? If it is the case, I agree that you have included previously the proper control to support your claim. 

Therefore, this manuscript is a demonstration that the previous con-seq is also capable to provide protection against the H2 strain. To my point of view, a short communication with the most relevant results will be more than enough. 

If the H2-con is a new and original con-sequence, as I have understood, you will need to include at least one WT control as requested. Without this comparison, you can not claim that the con-H2 seq is superior. Considering that the already described H1, H2, H3 and H5-con protein covers also the H2 subgroup, it would be very interesting to include it in the study to demonstrate the better performance of the new con-H2 protein to protect against H2 strains.

Author Response

Dear Reviewer 2,

Thank you for pointing out this discrepancy. We have included a statement in the methods that clarifies that the H2-con immunogenicity used in this study had been reported in our previous Sci Rep paper. The following statement has been included in the manuscript "This consensus H2 construct has been previously described and published by Lingel A., et al. [17]. In this study we have included a more detailed and comprehensive characterizion of the H2-con immunogen." (lines 177 - 179). All data in this current manuscript is new and has not been previously published. We hope that this eliminates the confusion on the origin of the H2-con and satisfies your request. Thank you for your time and efforts. Reviewers drive the momentum of science.

Sincerely,

Eric Weaver

Round 3

Reviewer 2 Report

OK for me then.